# WB-EMS Market Development—Perspectives and Threats

**DOI:** 10.3390/ijerph192114211

**Published:** 2022-10-31

**Authors:** Joshua Berger, Michael Fröhlich, Wolfgang Kemmler

**Affiliations:** 1German University for Prevention and Health Management, 66123 Saarbrücken, Germany; 2Department of Sports Science, Technische Universität Kaiserslautern, 67663 Kaiserslautern, Germany; 3Institute of Radiology, University Hospital Erlangen, 91052 Erlangen, Germany

**Keywords:** WB-EMS, electromyostimulation, EMS, home training, strength training

## Abstract

As a time-efficient and highly effective form of training, whole-body electromyostimulation (WB-EMS) enables personalised training for a wide range of users due to its personal training character and the individual control of the training intensity. However, due to misuse, negative side effects of WB-EMS have been reported in the past, resulting in expert guidelines for safe and effective WB-EMS application being issued. Furthermore, the use of WB-EMS is now legally permitted only for qualified personnel with certified equipment. This professionalization of the WB-EMS market as per the definition of quality standards for the devices and the personnel ensured a safe and effective WB-EMS application. However, recent market developments are undermining these standards through the growing of WB-EMS offers for the private sector. Hereby, most concepts focus on completely or predominately non-supervised WB application without control of potential overload by a qualified trainer. WB application is by no means trivial and the shift of responsibility for safety and effectiveness from the certified personnel to the trainees themselves is a clear step backwards in the development of WB-EMS use. We conclude that private, inadequately supervised WB-EMS application bears more dangers than potential benefits, not only for the trainees but also for the WB-EMS market as a whole.

## 1. Introduction

Whole-body electromyostimulation (WB-EMS) as a time-efficient, joint-friendly and highly effective training option enables individualized training for a wide range of users particularly due to its personal training character [1]. For novice applicants in particular, WB-EMS offers an easy way to start an effective and safe workout routine supported by individualized WB-EMS impulse intensity control and supervision of the voluntary exercises by experienced instructors [2]. Nevertheless, WB-EMS is also attractive for high-performance athletes and leads to significant improvements for them as well, since the involuntary EMS-induced contraction causes high metabolic and structural stress which is reported to result in significant improvements of parameters such as jumping power, maximum strength or shooting speed in soccer players [3,4]. Apart from specific (strength and power) outcomes, WB-EMS might also be beneficial for endurance athletes looking to prevent musculoskeletal complaints (e.g., low back pain) and injuries (e.g., knee lesions) with a time-efficient approach [5]. To emphasise this important aspect once again, so far, all effective studies conducted with WB-EMS have been closely supervised by qualified personnel [6]. This recognized concept is being challenged by new developments based on non-, or semi-supervised group or remote WB-EMS application.

## 2. Adverse Effects and Regulation of WB-EMS

In the wake of wide distribution of WB-EMS along with the initial lack of institutional regulation and standardization, several researchers have reported serious adverse effects and incidents after WB-EMS application. Since the DACH countries (i.e., Germany (Deutschland), Austria, Switzerland (Confoederatio Helvetica))–above all, the German WB-EMS market–are not only the largest [7] but also the earliest commercial EMS marketplaces, the developments and corresponding regulations that have arisen in Germany can be considered as a blueprint for future developments in other countries.

Especially in the early years of WB-EMS use, there were multiple adverse WB-EMS effects such as rhabdomyolysis, severe muscle pain and subsequent medical treatments which have since been addressed in scientific articles or in the public media [8]. Intensive or incorrect use of WB-EMS, e.g., by untrained personnel, can trigger these side effects after a single application and cause serious damage to the organism [9].

Most relevantly, in 2016, Malnick and colleagues published a statement on potential risks of WB-EMS, reporting several cases of WB-EMS-induced rhabdomyolysis. In one case, a 20-year-old male experienced rhabdomyolysis after a single application of WB-EMS. On the basis of this and further cases, the Israeli Ministry of Health published an official safety warning about WB-EMS and recommended that it should be used exclusively by healthcare professionals [10]. Slightly earlier, individual case studies on adverse WB-EMS effects were also published in the DACH area. Again, rhabdomyolysis occurred after a single or few WB-EMS applications, regardless of the physical fitness of the person [11]. In the study by Finsterer et al. (2015), it became quite obvious retrospectively that the patient was contraindicated to WB-EMS, which led to the call for a clear definition and consequent application of WB-EMS contraindications. These contraindications represent situations and medical conditions that preclude WB-EMS training either at a specific time, on a specific body part, without prior medical clarification (relative contraindications) or completely due to the potentially serious impact on health (absolute contraindications). [12]. In light of these rare adverse effects, in Germany, Austria and Israel great emphasis has been placed on safety, standardization, and monitoring aspects by the scientific WB-EMS community in DACH. Evidence-based recommendations, position papers and standards have been published [2] aiming to specify in-depth the application of WB-EMS in commercial, non-medical settings [13]. In particular, DIN 33961-5, a German norm on commercial non-medical WB-EMS facilities, provided requirements for equipment, operation, monitoring and application in adequate detail.

Nevertheless, the German Federal Ministry for the Environment, Nature Conservation, Nuclear Safety and Consumer Protection (BMU) launched a regulation on the application of non-ionizing radiation to humans (NiSV) in 2018, which includes WB-EMS application in its commercial application [14]. Apart from accurate recording and official monitoring of registration, individual training aspects, adverse effects and function of the WB-EMS unit, the most significant innovation of the NISV is the mandatory education and certification (“Fachkunde EMF-S”) of the instructor conducting and supervising the WB-EMS application. After COVID-19 related delays, the Fachkunde EMF-S will come into force on 1 January 2023 and will have a major impact on the German WB-EMS market. Unfortunately, despite draconian sanctions (€50,000), the majority of WB-EMS facilities in Germany have not adequately implemented (or even realised) this mandatory requirement, so that the timely qualification of the approximately 5000 WB-EMS instructors will be difficult to accomplish by the deadline at the beginning of 2023.

## 3. New Development on WB-EMS Settings—Problems and Possible Consequences

In response to the upcoming regulation and Fachkunde EMF-S with its inconvenience and costs, the COVID-19-induced lockdown of WB-EMS facilities in Germany, and the “Peloton” phenomenon, new business models are focusing on less- or completely non-supervised WB-EMS settings. A not entirely new concept here is group WB-EMS application with (if at all) one instructor supervising a multitude of applicants. This might be an effective and safe concept for “spinning” programs, bearing in mind that remote monitoring of vital parameters (e.g., heart rate or blood pressure) by the instructor is a common practice nowadays. However, monitoring and regulation of vital parameters to control the intensity related to WB-EMS is less simple. In this context, the most crucial issue is the (subjective) regulation of the impulse intensity that requires a close interaction and close distance between instructor and participant [15,16]. This includes (1) frequent feedback from the participant about perceived exertion for each area of stimulation, (2) permanent visual monitoring of the participant and eye contact to check participant stress, avoid overload and to react immediately to the first signs of cardiorespiratory or metabolic side effects, and (3) verbal and haptic movement corrections and rapid assistance in cases of emergency, particularly cutting off the device power supply and preventing fall-related injuries [1]. Another difference to the predominately acute adverse effects of exercises eligible for group fitness concepts lies in the fact that rhabdomyolysis, as a highly relevant side effect of WB-EMS-induced overstressing, did not peak before day 3–4 post-exercise [9], indicating once again the crucial role of the experienced and well-trained instructor. This leads us to strongly advise a 1:1 instructor–participant ratio, although a 1:2 ratio is also considered tolerable for non-medical WB-EMS application with less critical participants [2].

A more recent approach is home WB-EMS settings that range from a small monthly rental fee for equipment and contents via apps or a hybrid system of supervised on-site training to mobile training for “on the go” and additional home training. With respect to non-supervised concepts, DIN 33961-5 and in particular the German radiation protection committee (SSK) excluded the private, non-supervised application of WB-EMS. However, as one of the most crucial limitations of the mandatory NISV, launched primarily to increase the safety of (among others) WB-EMS, there is no official ban on this setting yet. However, common sense suggests that excessive non-supervised private WB-EMS application will significantly increase the prevalence of adverse effects; in fact, an online survey [17] on individual non-supervised home training programs during the COVID-19 pandemic indicated a large number of adverse effects. Of the 1902 respondents, 12% stated that they had suffered a musculoskeletal injury during their exercise training. The authors suggested the occurrence of these injuries may be due to intensive/frequent training and a lack of individualization of the training program, whether self-designed or recommended online by social media influencers.

The independent application and thus the shift of responsibility from certified personnel to users personally would therefore be a clear step backwards in the development of WB-EMS training and hold more dangers than potential benefits, both individually for the trainees and for the entire industry. Apart from the safety aspects, all of these aspects prevent an optimum training effect independent of the outcome. As already discussed, this is all the more the situation for WB-EMS with its high demands regarding individualization. Thus, bypassing the attentive and experienced instructor and imposing the task of ensuring safety and effectiveness on the trainees should be considered a “no go” in commercial WB-EMS application. However, even banning non-supervised commercial WB-EMS will still leave the critical issue of private WB-EMS application at home. The latter also includes the negative aspect of low quality WB-EMS devices not being subjected to the German medical device act, used while ignoring or neglecting the contraindications of WB-EMS.

The question arises: Is there a way to rate the role of remote WB-EMS programs with consistent online supervision by a certified instructor? One may argue that this depends greatly on whether group or personal training settings are being applied. We agree so far that, of course, a lower instructor–participant ratio would increase the individualization of the application. Technical innovations might allow the instructor to monitor or even regulate training parameters during the session and prevent excessively frequent application (i.e., >2 sessions per week [2]) and hence solve, if desired, some of the safety and effectiveness issues of WB-EMS discussed above. However, this does not alleviate the common problems of remote application even when applying a 1–1 setting: the insufficient face-to-face monitoring, the inability to apply haptic corrections and, in particular, the absence of rapid assistance in cases of emergency.

## 4. Consequences of Unrestricted WB-EMS Application and Undermining the NISV

The current developments in WB-EMS group training and supervised and non-supervised home use raise the question of whether these concepts will be a helpful next step for the fitness market or whether they will backfire. We might appear pessimistic, but we do feel that the latter will become true. Without further restrictions, inadequate WB-EMS concepts launched by providers without any interest in the sustainable development of WB-EMS will increase the prevalence of adverse effects, perhaps even fatalities. Given that the radiation protection commission classifies WB-EMS as a high EMF exposure, this puts commercial WB-EMS only a hair’s breadth away from exclusive medical use. Apart from the desirable ban on private WB-EMS use, further consequences to regulate WB-EMS could be enforced by the legislator. According to a recent consensus conference of the EMS Expert Group Germany, medical WB-EMS training is a (1) primarily therapeutic intervention (2) based on an existing diagnosis (3) that is provided by qualified medical–therapeutic personnel (4) in compliance with current guidelines and (5) using medical devices. Therefore, the consequences of exclusive medical WB-EMS use in commercial facilities will definitely be a threat to the existence of the roughly 2000–2500 WB-EMS studios in DACH and the role of WB-EMS as an efficient personal training setting. Even more harmful is the fact that future serious adverse effects might permanently damage the image of WB-EMS as an individualized, safe, and effective training for a large user cohort. Thus, when considering the ongoing technical progress, we feel that the WB-EMS training should not be oriented exclusively to what is technically feasible, but to what makes sense from a training science point of view.

## 5. Conclusions

To date, WB-EMS can be considered as a safe and effective technology that can be applied as time-efficient, joint-friendly and individualized training in health and performance. This favorable development is based on expert recommendations, norms and mandatory regulations. The undermining of this status by non- or inadequately supervised EMS settings should be considered as a step backwards in the development of WB-EMS. We hope we are wrong, but we feel that further serious adverse effects will result in drastic responses by regulatory authorities with a significant impact on the German WB-EMS market. One may argue that this is a specifically German issue; on the other hand, technical innovations, new business models for home training and the risk of upcoming pandemic lockdowns are worldwide scenarios and hence similar developments will in time affect all the other countries which have relevant WB-EMS applications.

## Data Availability

Not applicable.

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
