# Peer review of "WB-EMS Market Development—Perspectives and Threats"

_ijerph, 2022, doi:10.3390/ijerph192114211_

Round 1

Reviewer 1 Report

I thank the Editors for the opportunity to review the manuscript entitled “WB-EMS Market Development - Perspectives and Threats”.

Berger et al. describe in their Opinion Paper the recent WB-EMS market developments and discuss potential pitfalls regarding the increasing trend of unsupervised home-based WB-EMS exercise.

Given the growing popularity of more time-efficient exercise modalities including WB-EMS, the authors should be commended for their critical presentation of the potential benefits and hazards (when used improperly) of WB-EMS, which could be of interest for those working in the field.

The manuscript is generally very well written and clearly structured throughout. However, some issues remain to be addressed before this paper may be considered for publication.

General/major comments:

1) Several statements require a more careful citation (in detail, see below in specific comments).

2) The paper would greatly benefit from a more detailed characterization of critical participants, including, in particular patients and individuals with pre-existing conditions/risk factors (including appropriate references).

Specific comments:

Lines 27-29: Please include proper references (particularly regarding safety)

Line 40: What about WB-EMS application for other health conditions? This topic is getting increasingly important. Authors should discuss whether WB-EMS can be regarded feasible for clinical populations or should it exclusively be used in athletes and recreational athletes. In this context, it is important to present contraindications and limitations of WB-EMS.

Moreover, advantages and disadvantages compared to other more traditional exercise options should be discussed (e.g. classical resistance training)

Lines 52-54: Authors only discuss rhabdomyolysis as adverse WB-EMS effect. Surely on of the major AEs associated with WB-WMS but for a more comprehensive overview, I suggest to present some more potential AEs as you state that a considerable number of AEs have occurred (lines 52-53). Here, some prevalence numbers should be included (in order to be able to better classify the statement "considerable number")

Line 64: What kind of contraindications?

Line 76: Are there specific regulations regarding WB-EMS application in clinical settings (compared to commercial application)?

Line 108: Reference is needed to support this statement.

Line 121: Is this a result of the survey or just an assumption of the authors?

Lines 155-156: Sounds a little bit polemic. I recommend to revise this sentence.

Line 159: "[...] most inexpensive personal training setting". This statement sounds very daring and I am not convinced of it. There are many options in personal training to work with little or even no additional equipment. Therefore, I recommend to revise this sentence or to include a powerful reference to support this statement.

Reviewer 2 Report

Comments to Author

This is an opinion piece which focuses on the regulation of whole body electromyostimulation training (WB-EMS) which was interesting to read. This technology is becoming more available in countries around the world so has relevance. The manuscript covers some of the issues, but the language and focus could be clarified. I feel more insight could be given in a number of areas and I have suggested some ideas below. Also there are many sentences which are long and confusing with lots of commas. For example Line 113-119 – poor structure, Line 136 poor grammar example. Grammar example… “A not entirely new concept” (page 2 line 90).

The manuscript really only offers one likely solution to this problem of adverse events in depth. Could there be other solutions besides regulation? 

Are there any benefits of not having such stringent regulation? What are the pros to having people use these devices at home? Perhaps some people will gain health benefits that they would not have otherwise if forced to go to a class or pay for a trainer that they cannot afford.

Is there an issue with the application of WB-EMS to clinical populations who have reduced sensation or other issues; this wasn’t discussed. These persons may be at heighten risk and may be more desperate to “find a cure”.

Please define acronyms, i,e, What does SSK stand for? What does NISV stand for?

Sections

1. Introduction

References are sometimes missing after an important statement. I feel the effectiveness of WB-EMS is overstated. Your reference 6 suggests that WB-EMS needs a lot more investigation and current evidence doesn't suggest it is that effective. This reference was quite informative. I think you should talk more about the safety and less about how effective you believe it is. Effective or not ... that wont hurt anyone but safety problems can.

2. Adverse effects and regulation of WB EMS

More specific information is needed in this section. Do you have list of other forms of injury besides Rhabdo? How common are these adverse events? How does it compare to the number of adverse events experienced by people exercising at home by themselves in non EMS activities?.

Will the rest of the world accept in the same way? Perhaps this would come under physio therapist or exercise science in some countries … is this likely in Europe? It might be good to consider one of these bodies regulating this technology or the qualification.

This manuscript is also a chance to educate about Rhabdo and the risks from WB-EMS; I think if you want to prevent overregulation this might be a good idea. Voluntary eccentric exercise for example has a great amount of potential for Rhabdo. What is the incidence of exertional rhabdomyolysis? Has it increased lately? More info could be given about those most at risk of Rhabdo (maybe things like male?, non-elite?, regular exercisers overweight or very muscular?). Plus some info about the training chance of developing rhabdomyolysis (multiple muscle groups, extreme temperatures, not hydrating properly?) and whether some medications increase the risk.

3. New development on WB-EMS settings

line 93… One does not monitor HRmax, I think you mean HR or %HRmax. Also respiration rate and oxygen saturation are probably less important than blood pressure. Oxygen saturation would really only be measured in cardiopulmonary populations. Maybe you should make a comment on what they should be measuring for safety.

What screening is required for WB EMS? Is it similar to a pre-exercise screen?

What are the requirements for the Din 33961 – 5?

What does getting the Fachkunde EMF-S require and why have the majority of facilities not implemented the mandatory requirements… Are there some sticking points?

What remote-control does the trainer have over each person stimulation. Maybe this needs to be improved? Surely systems can be programmed to limit the stimulation intensities.

4. Consequences of unrestricted...

line 159 why do you say WB EMS is the most inexpensive training setting?
